# Meter-scale heterostructure printing for high-toughness fiber electrodes in intelligent digital apparel

Gun-Hee Lee [1,2,14], Yunheum Lee [3,14], Hyeonyeob Seo [3,14], Kyunghyun Jo [4], Jinwook Yeo [5], Semin Kim [6], Jae-Young Bae [7], Chul Kim [3], Carmel Majidi [8], Jiheong Kang [9], Seung-Kyun Kang [7], Seunghwa Ryu [5] & Seongjun Park [1,10,11,12,13] ✉

Intelligent digital apparel, which integrates electronic functionalities into clothing, represents the future of healthcare and ubiquitous control in wearable devices. Realizing such apparel necessitates developing meter-scale conductive fibers with high toughness, conductivity, stable conductance under deformation, and mechanical durability. In this study, we present a heterostructure printing method capable of producing meter-scale (~50 m) biphasic conductive fibers that meet these criteria. Our approach involves encapsulating deformable liquid metal particles (LMPs) within a functionalized thermoplastic polyurethane matrix. This encapsulation induces in situ assembly of LMPs during fiber formation, creating a heterostructure that seamlessly integrates the matrix's durability with the LMPs' superior electrical performance. Unlike rigid conductive materials, deformable LMPs offer stretchability and toughness with a low gauge factor. Through precise twisting using an engineered annealing machine, multiple fiber strands are transformed into robust, electrically stable meter-scale electrodes. This advancement enhances their practicality in various intelligent digital apparel applications, such as stretchable displays, wearable healthcare systems, and digital controls.

The integration of digital functionality into textiles has garnered significant attention, particularly with the increasing demand for wearable healthcare systems, smart interactive systems, and human-computer interfaces[1-7]. Realizing the vision of digital smart textiles requires the development of fiber-type electrodes with specific characteristics: 1) deformability, 2) high toughness, 3) stable conductance, and 4) scalability for mass production[4]. However, traditional metal wire-based fiber electrodes struggle with compatibility with the flexible human body and often suffer fatigue fractures during prolonged use[4]. In response, many research efforts have focused on developing soft, stretchable, conductive fibers with high toughness. Nevertheless, conventional rigid conductive filler-based fibers show a high gauge factor under strain, rendering them unsuitable for use as electrodes or interconnects[5,8-10]. Furthermore, the coating process of conductive fillers on the fiber surface is typically restricted to a lab-scale production, limiting its suitability for large-scale manufacturing—a crucial aspect for actualizing smart wearable clothing[8,11].

Recently, there has been a growing interest in integrating gallium-based liquid metal (LM) into stretchable fibers because of their ability to retain electrical conductance even under extensive deformation[12-18]. However, the fluidic nature of LM leads to challenges such as instability and leakage, highlighting the need of effective encapsulation (Supplementary Fig. 1, Supplementary table 1)[15-17]. Challenges emerge when trying to expose the LM for electrical connections by stripping the

outer layer, as this can lead to immediate leakage (Supplementary Fig. 2). Consequently, the practical use of this technology has largely been limited to heaters and triboelectric generators, rather than as fiber-type electrodes or interconnects[15–17]. In summary, currently, there is no fiber electrodes satisfying all the requirements for stretchability, high toughness, low gauge factor, and scalability for mass production.

In this study, we present a scalable heterostructure printing method for the mass-producing tough and stretchable conductive fibers (TSF) using LM particle (LMP)-embedded thermoplastic polyurethane (TPU) (Fig. 1a, Supplementary Fig. 3). The fabrication of large-scale, self-supporting TSF involves the development of a biphasic heterostructure, which merges a leakage-free conductive LMP region with a highly tough TPU backbone. This is achieved through the in-situ assembly of chemically activated TPU that binds to the bridged LMP. Additionally, we've developed a process that involves twisting and sintering multiple strands of TSF to further enhance toughness and ensure stable conductance, resulting in the production of twisted TSFs (TSF$^{tw}$), with high robustness and fidelity (Fig. 1b and Supplementary

Fig. 4). Consequently, our TSF$^{tw}$ can be seamlessly integrated into clothing, alongside various electronic components, facilitating the creation of wearable healthcare systems, digital control units, and interactive displays (Fig. 1b).

A crucial feature of our design is the prevention of LM leakage during usage. As illustrated in Fig. 1c, both sides of the TSF are enveloped with TPU, effectively preventing LM leakage from the top and bottom surfaces of the fiber when under strain. The surface of the conductive region is protected by a TPU layer, evident in the SEM image, further enhancing the mechanical stability of the TSF while maintaining high conductivity. Moreover, we fabricated large-scale TSF$^{tw}$ by twisting multiple strands of TSFs using a coiling machine (Fig. 1d and Supplementary Fig. 5) while this substantial length significantly enhances its applicability in human-sized digital apparel. The large-scale TSF$^{tw}$ also exhibits remarkable electrical conductivity and maintains stable conductance even upon stretching (Fig. 1e and Supplementary Video 1). Figure 1f provides a comparative analysis that highlights the superiority of our TSF in terms of electrical conductivity,

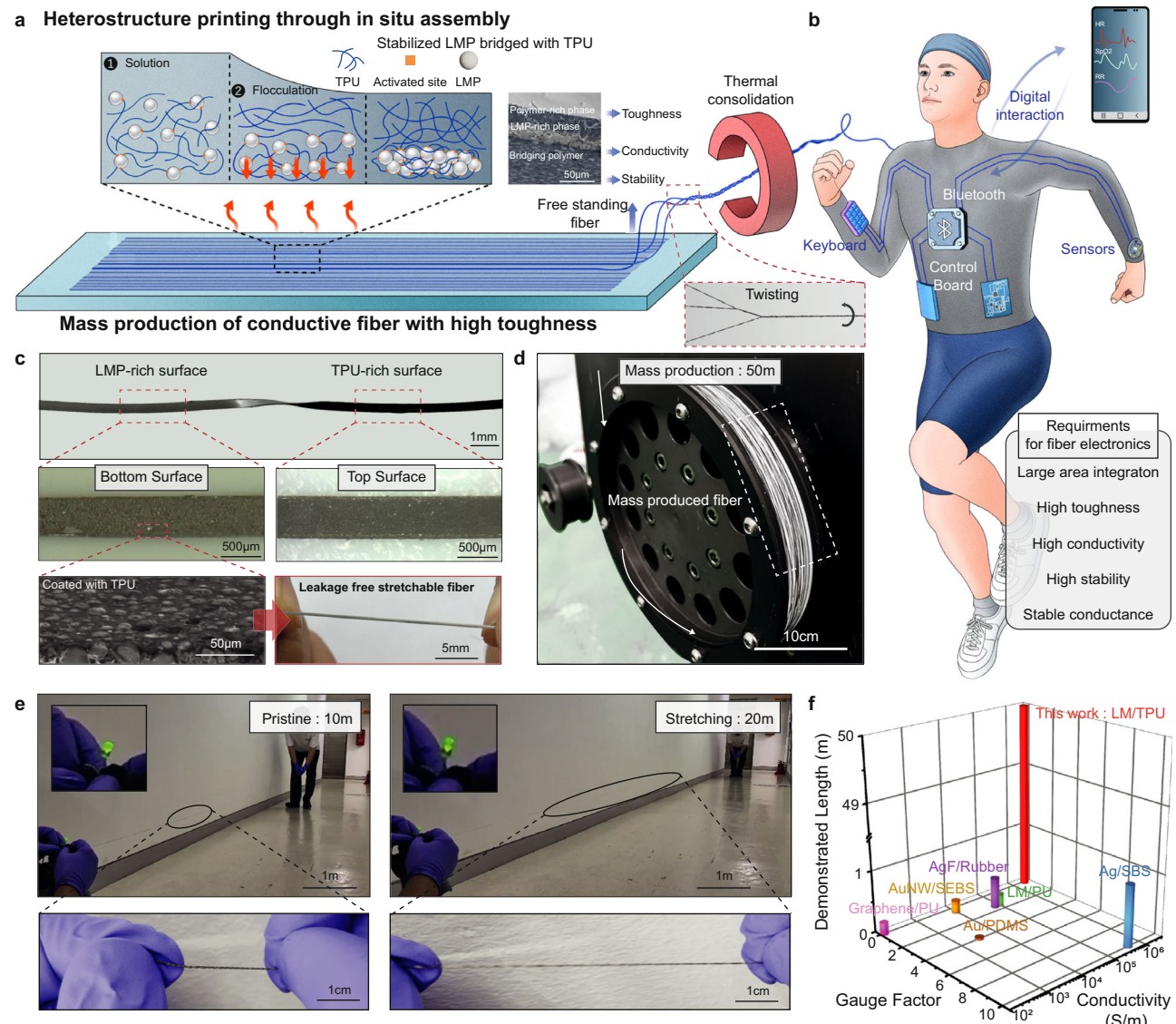

**Fig. 1 | Tough and stretchable conductive fiber (TSF). a** Schematic illustration of the mass fabrication process for hetero-structured tough, stretchable, and conductive fiber (TSF) and twisted TSF (TSF$^{tw}$). **b** Schematic illustration of wearable digital system fabricated with TSF$^{tw}$. **c** Photograph, OM, SEM image of tough and stretchable fiber without leakage of LM. Photograph of TSF with top and bottom surface. There are no notable leakages of liquid metal under strain. **d** Photograph of mass-produced twisted TSF. **e** Photograph of 10 m-long twisted TSF before and after stretching 100%. Inset is LED connected to source meter through twisted TSF. **f** Comparison with demonstrated length, Gauge factor, and initial conductivity of conventional stretchable fiber-type electrodes.

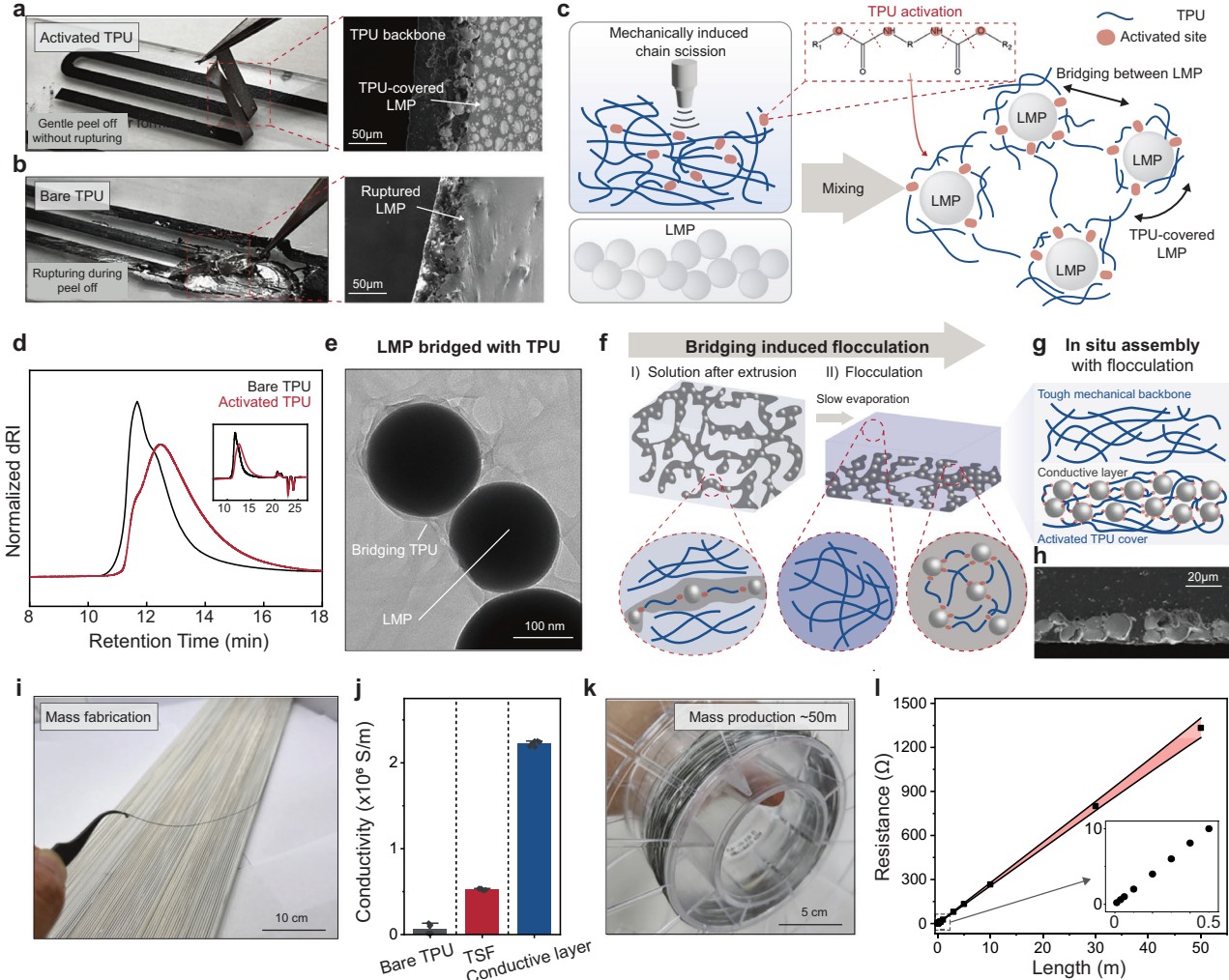

**Fig. 2 | Solution process for mass production of free-standing, hetero-structured tough, stretchable, and conductive fiber (TSF). a, b** Photograph of TSF peeling off process and cross-sectional SEM images of activated TPU (a) and bare TPU (b). **c** Schematic illustration of activation of TPU and LMP bridging. **d** Gel permeation chromatography of TPU before (black) and after tip sonication (red). **e** A TEM image of LMPs bridged with activated TPU. **f** Schematic illustration of flocculation due to bridging of LMPs. **g** Schematic illustration of cross-section in situ assembled in three layers. **h** Cross-sectional SEM image of in situ assembled TSF. **i** Photograph of fabricated free-standing TSF. **j** Conductivity of fiber fabricated with bare and activated TPU. Values represent the mean and the 1.5 IQR (*n* = 6). **k** Photograph of mass-produced TSF with activated TPU. **l** Resistance of TSF along with the length. Error bars indicate the standard deviations (*n* = 3).

a low gauge factor, and exceptional scalability relative to other fiber-type electrodes (Supplementary Fig. 6, Supplementary table 2), whose traits are foundational for advancing digital clothing[19–24].

## Results

### Large-scale fabrication of hetero-structured free-standing TSF

To create free-standing TSF, we employed commercially available TPU as the foundational matrix and covering polymer for LMP. This choice was driven by TPU's exceptional toughness and polarity (Supplementary Fig. 7). Additionally, we opted for NMP as the solvent due to its slow evaporation rate, ensuring uniform fiber formation at a large-scale compared to volatile solvents that evaporate rapidly.

The process of crafting free-standing TSF involves activated TPU with high-power tip-sonication (Supplementary Fig. 8), resulting in heterostructure fiber formation without rupturing (Fig. 2a). SEM images illustrate the bottom LMP covered with TPU, ensuring a leakage-free and mechanically stable conductive fiber. Conversely, non-activated TPU leaves the bottom LMP uncovered, causing severe rupturing during peel-off and rendering free-standing fiber unfeasible with bare TPU (Fig. 2b). As shown in Fig. 2c, treatment with high-power sonication breaks the bond within the TPU. The activated TPU, now

featuring chemical functional groups, covers the LMP and facilitates the formation of bridges among LMPs during the mixing process. Gel permeation chromatography (GPC) profile in Fig. 2d confirm the presence of a distinct segment characterized by a diminished molecular weight of TPU subsequent to the application of sonication. The profile can be deduced that the functional groups are formed at the end of the TPU activated by sonication, which have a strong affinity for the gallium oxide present on the surface of the LMPs[25,26]. The proton nuclear magnetic resonance (1H NMR) spectrum presented in Supplementary Fig. 9 implies the occurrence of amine group formation in TPU activated through sonication. The H signals, a triplet at 1.22–1.26 p.p.m. and a singlet at 1.67 p.p.m. adjacent to the amine group, serve as evidence of peptide bond cleavage and simultaneous amine group formation. The activated TPU efficiently encapsulates the LMPs and forms bridges between particles, as evidenced by the TEM image (Fig. 2e) and TEM energy dispersive X-ray spectroscopy (EDS) (Supplementary Fig. 10).

For the reliable mass fabrication, the viscosity of solution is important (the viscosity of solution is presented in Supplementary Fig. 11). A low-viscosity solution, when spread after extrusion, is incapable of producing a robust and thick free-standing fiber. In

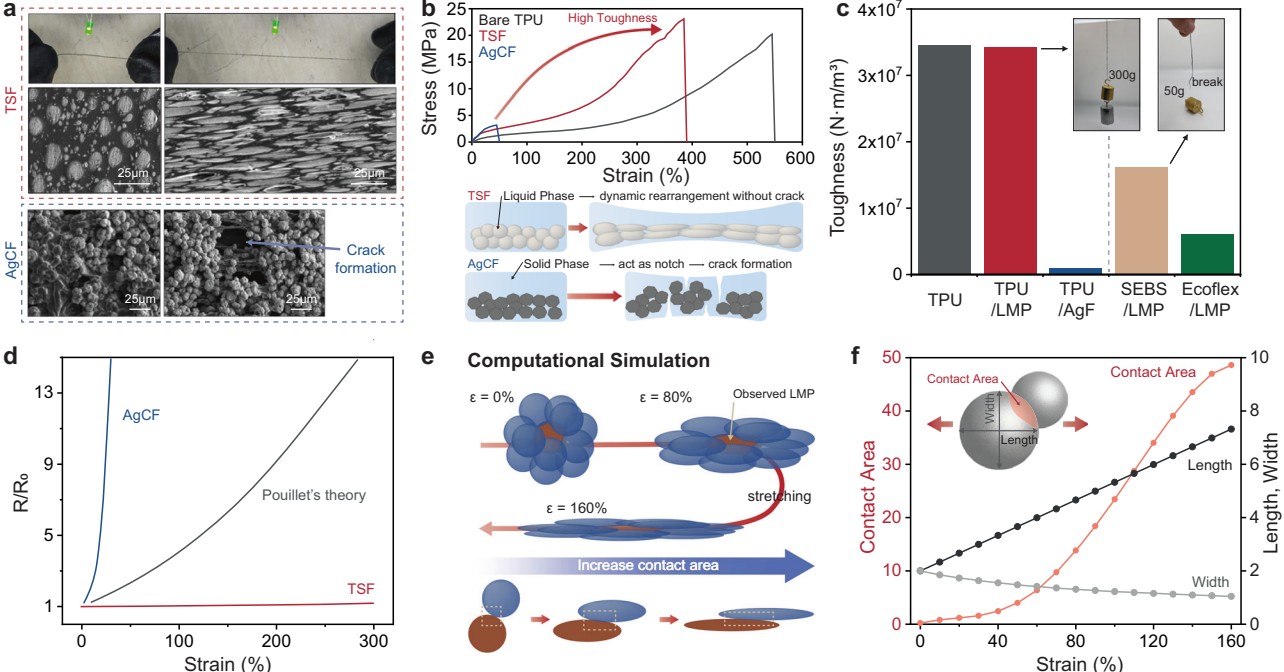

**Fig. 3 | Toughness, stretchability, and stable conductance of tough, stretchable, and conductive fiber (TSF). a** Photograph and SEM image of TSF and silver flake embedded fiber (AgCF) before and after stretching. **b** Stress-strain curve of TSF and AgCF. **c** The toughness of TSF and comparison with various stretching fibers. **d** Relative resistance variation of TSF and AgCF according to strain. **e** Schematic illustration of computational simulation regarding LMP elongation under strain. **f** Computational simulation regarding geometrical variations of LMP under strain.

contrast, a high-viscosity solution maintains its thickness, making it suitable for the fabricating free-standing fibers and films. However, the issue with the high-viscosity solution with bare TPU is that LMPs do not undergo phase separation (Supplementary Fig. 12)[27], which results in the film that lacks both electrical conductivity and mechanical toughness (Supplementary Fig. 13). To achieve the desired combination of toughness and conductivity in the fiber, it is imperative to establish phase separation between the tough backbone and conductive regions. Here, we found that LMP covered with activated TPU, phase separation in the high-viscosity solution is achieved. The bridging between particles establishes physical connections and forms larger clusters (Supplementary Fig. 14), resulting in enforced phase separation, referred to as flocculation (Fig. 2f)[28,29]. The flocculation is driven by the bridging effect of activated TPU, in which terminal amine groups on TPU chains form interparticle linkages between adjacent LMPs, effectively inducing LMP clusters that promotes phase separation even in a high-viscosity solution. These heavy LMP clusters are pulled downward by gravitational force, creating an electrically conductive region, while the TPU separates upward, forming a robust mechanical backbone that imparts stretchability and toughness to the film (Fig. 2g). The separated Fig. 2h and Supplementary Fig. 15 present a cross-sectional SEM image of the hetero-structured TSF. The SEM image visualizes the densely packed LMP region at the bottom, which facilitates the formation of a conductive pathway through strain-induced elongation and connection of the particles. Furthermore, this phenomenon occurs across a range of solution viscosities, enabling precise tuning of fiber thickness (Supplementary Fig. 16).

With the designed experimental setup, we can reliably fabricate free-standing fibers or films for the mass production of TSF (Fig. 2i). Only TSF with activated TPU undergoes the necessary phase separation, resulting in the electrical conductivity (Fig. 2j). During fiber peel-off for free-standing applications, the LMP is electrically connected to the localized strain (Supplementary Fig. 17, Supplementary Video 2). The slow evaporation of our solvent, NMP, enables the very uniform and dependable fabrication of TSF on a large-scale (a uniformity of TSF is presented in Supplementary Fig. 18), reaching lengths of approximately 50 meters (Fig. 2k). The reliable electrical connection in our fiber allows for a linear increase in electrical conductivity up to 50 meters (Fig. 2l).

## Mechanical and electrical properties of TSF

The characteristics of TSF, including stretchability, high toughness, and electrical conductivity, are prominently demonstrated in the LED connection, which exhibits continuous operation even during extensive stretching (Fig. 3a)[30]. The scanning electron microscope (SEM) image of TSF reveals that LMP elongates under strain without any signs of delamination from the polymer matrix[31–33]. In contrast, conventional conductive fibers embedded with solid silver flakes (AgCF) are prone to delaminate from the polymer and form cracks within the polymer matrix during stretching[34]. This behavior contrasts with that of TSF, which shows superior stretching capabilities and toughness (Fig. 3b). We attribute this difference in mechanical performance to the presence or absence of a stiff filler. In fibers incorporating a stiff solid filler, the mismatch in elastic modulus induces stress concentration, facilitating crack initiation and propagation, thereby compromising mechanical toughness[35]. In contrast, in TSF, all additives exhibit deformability comparable to that of the polymer matrix, preventing the formation of voids or microholes that could act as stress concentrators within the elastomer structure. Furthermore, the conductive LMP-rich region is spatially segregated from the tough polymer matrix, as illustrated in the subsequent figure. Consequently, TSF retains mechanical properties comparable to those of pure TPU fiber, ensuring both structural integrity and functional performance.

For practical fiber applications, toughness is a critical factor to consider[36]. The high toughness of TPU makes it highly suitable for functional clothing, and it is already a commercialized material, as seen in the case of Spandex[37]. As shown in Fig. 3c, the toughness of our fiber is similar to that of bare TPU, indicating its potential for use in conventional commercial systems. In contrast, AgCF exhibits much lower toughness than TSF, rendering it unsuitable to smart clothing

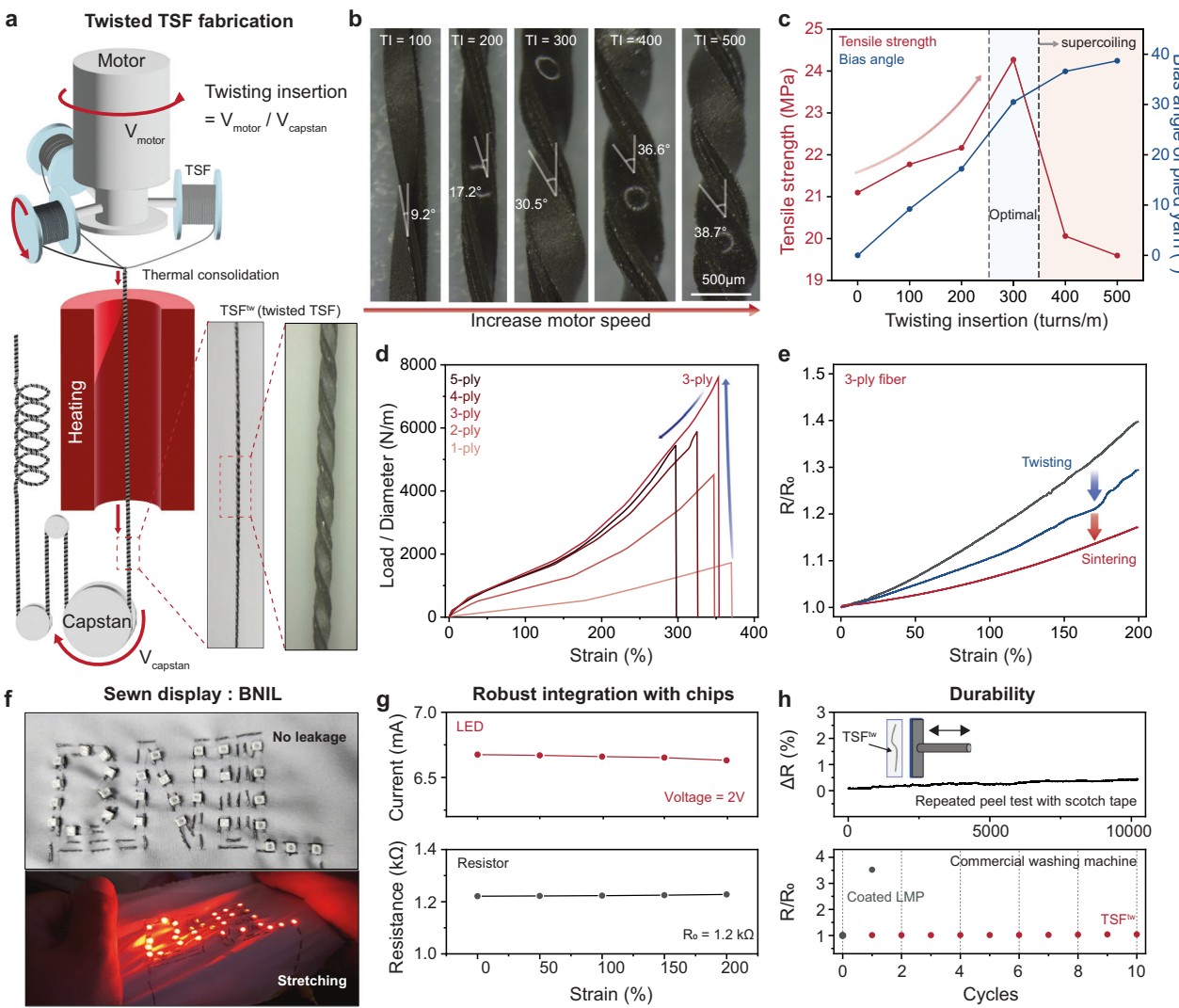

**Fig. 4 | Fabrication strategies and mechanical, electrical properties of twisted TSFs. a** Schematic illustration of mass fabrication of twisted TSFs. **b** Optical microscopy image of twisted TSFs according to twisting insertion. **c** Tensile strength and bias angle of twist TSFs according to twisting insertion. **d** Normalized load under application of strain according to a number of twisted fibers. **e** Relative resistance under strain of pristine TSFs, twisted TSFs, and twisted TSFs after sintering (TSF^tw). **f** Photograph of TSF^tw-sewn deformable textile display. **g** TSF^tw-integrated electronic components (LED and resistor) operation under strain. **h** Mechanical durability and washability of TSF^tw.

applications. Other fibers containing LMP and elastomers such as Styrene-Ethylene-Butylene-Styrene (SEBS) or ecoflex also exhibit lower toughness, making them less suitable for practical use due to both their insufficient toughness and difficulties in the large-scale fabrication process (Supplementary Fig. 19).

To ensure the reliable operation of stretchable electronics, maintaining stable conductance under strain is crucial[10,38,39]. However, conventional rigid stretchable fibers based on conductive fillers tend to experience a significant reduction in electrical conductivity when subjected to strain[8,10,40,41]. This reduction occurs because of the percolation of conductive nano/micro fillers, and when strain is applied, the physical contact between these conductive fillers is disrupted, resulting in a dramatic increase in resistance[8,42]. This increase is more significant than what is predicted by Pouillet's theory, as shown in Fig. 3d. In contrast, when TSF is subjected to strain, the change in resistance is much smaller than theoretically expected. We hypothesize that this behavior arises from a competition between the geometrical elongation of each particle and an increase in the contact area between particles under strain (Fig. 3e) and the SEM image[30]. To confirm our hypothesis, we conducted computational simulation (Detail

description on simulation is presented in Supplementary Fig. 20), and the simulation results demonstrate a linear increase in length along the loading direction and a decrease in width, contributing to increased resistance. However, the contact area between particles, which is directly related to resistance of percolation network, exhibits a faster nonlinear growth, effectively alleviating the rise in resistance (Fig. 3f)[43]. Furthermore, the activated TPU encapsulates the LMP, effectively preventing particle leakage or rupture under high-strain conditions (Supplementary Fig. 21).

## Enhanced mechanical and electrical properties of TSF^tw

The technique of twisting multiple fibers is a well-established method employed to enhance the toughness of fabrics[44–47]. In our case, we have adopted this approach to our TSF to fabricate large-scale of TSF^tw with further augment its mechanical robustness as well as electrical stability. To precisely control fiber twisting and thermal annealing in the manufacturing process, we devised a designed annealing machine (Fig. 4a). Using this system, we were able to fabricate fibers with controlled twisting insertion (TI) by modulating both motor speed and capstan speed (Fig. 4b), which is highly important to enhance the

mechanical robustness of fiber[48,49]. Here, tensile strength increased up to 300 TI due to increased fiber friction, but beyond this threshold, leading to a decline in strength as the bias angle of the fiber increased (Fig. 4c and Supplementary Fig. 22). Furthermore, supercoiling of fiber was observed above 400 TI, making it impractical to usage in fiber-integrated devices (Supplementary Fig. 23). Therefore, we have designated 300 TI as the optimal point for our TSF[tw]. Subsequently, we conducted a comprehensive evaluation of the TSF[tw] properties concerning varying ply condition. Our experimental results confirm that a three-ply configuration of TSF is the optimal design, finding that increasing the ply count beyond four does not result in further enhancements in toughness (Fig. 4d). Here, we note that smart clothing applications primarily rely on body deformation, which is typically less than 200%[50]. Therefore, our focus is on the strain region below this threshold. Regarding electrical performance, TSF[tw] exhibits a lower gauge factor compared to a single TSF, as demonstrated in Fig. 4e. This reduction primarily stems from the increased effective width of the conductor resulting from the twisting and sintering processes applied to the TSFs. Additionally, heat treatment promotes a more intimate connection between the conductive layers of fibers, contributing further to the reduction in the gauge factor. Our findings indicate that the resistance variation in micro-scale fibers is notably influenced by their effective width, considering the non-negligible scale of our conductive filler, micro-scale LMP. Detailed explanations are presented in Supplementary Figs. 24 and 25.

The enhanced toughness of TSF[tw] prevents unwanted elongation after strain. When subjected to strain, TSF exhibits elongation afterward, whereas TSF[tw] does not show significant elongation after stretching and maintains its mechanical properties even after undergoing 10,000 repeated cycles (Supplementary Fig. 26). Similarly, the electrical resistance of TSF[tw] is maintained over 10,000 cycles of strain, exhibiting consistent behavior under strain before and after the strain cycles (Supplementary Fig. 27).

The mechanical stability and toughness of TSF[tw] make it easily attachable to conventional textiles through suturing (Fig. 4f)[51–53]. Moreover, unlike conventional sheath-core structure fibers fabricated through injection, which require additional cut-and-paste steps, our TSF[tw], as a fiber-type electrode, can be directly interconnected with conventional chips (Supplementary Fig. 28). This robust integration with chips ensures stable operation under strain, as evident in the bottom image of Fig. 4f. Since the electrical conductance of TSF[tw] remains stable even under strain, electronic components interconnected with TSF[tw] maintain stable operation even under stretching conditions (Fig. 4g)[13]. Finally, we have confirmed the mechanical durability of TSF[tw] through repeated peel tests with Scotch tape (Supplementary Fig. 29) and washing with commercial machine, demonstrating that our TSF[tw] maintains stable electrical properties even when subjected to external stimuli, which facilitates the realization of practical digital wearable systems (Fig. 4h)[54].

### Intelligent digital apparel with TSF[tw]

The large-scale fabrication capability, mechanical stability, high toughness, and stable conductance of TSF[tw] enable the realization of smart clothing through the integration of electronic components and circuits onto commercial garments. Figure 5a shows a photograph of an intelligent digital apparel (IDA) fabricated using eight 50-cm long TSF[tw] strands. The IDA comprises a controller, a Bluetooth communication module, an interactive display, a keyboard, and a photoplethysmography (PPG) sensor, all connected through TSF[tw] interconnects and electrodes (Fig. 5b and Supplementary Fig. 30). The toughness and thinness of the TSF[tw] enable suturing with a standard commercial needle (Supplementary Fig. 31), and tight integration with circuit boards with a tying method (Supplementary Fig. 32). To further prevent unwanted electrical shorting or possible leakage, we also developed and employed a fast, on-the-fly

encapsulation method for TSF[tw] using SEBS (Supplementary Figs. 33 and 34).

Since the electrical conductance of TSF[tw] is maintained even under mechanical deformation and elongation, the electrical components on the IDA continue to operate even during body motion-induced deformation and vigorous exercise (Fig. 5c and Supplementary Fig. 35 and Supplementary Video 3). Furthermore, all the light-weight components and the digital systems for wireless communications on the IDA also enables the untethered usage in daily life, as illustrated in Fig. 5d. In addition, mechanical stimuli applied to the IDA during daily life do not damage the TSF[tw] or the integrated system (Supplementary Fig. 36). One of the primary needs in smart clothing is the ability to monitor health states and facilitate digital interaction[55]. We have demonstrated these two essential features with TSF[tw]-used smart clothing. PPG is an effective method for monitoring heart rate, providing valuable insights into a person's physical condition or detecting abnormal heart activity that may require urgent medical attention[56]. Fig. 5e displays real-time PPG data acquired with our IDA. This reliable monitoring capability remains effective even during arm movements involving stretching and bending, and even the heart activity before and after a workout could be monitored by our IDA (Supplementary Fig. 37).

Furthermore, we have provided a comprehensive overview of the wireless digital interactive system based on the IDA, including detailed circuit diagrams, Bluetooth codes, and the keyboard operation mechanism (Supplementary Fig. 38). With this digital interaction system on the IDA, we showcased its functionality by playing a maze-solving game (Fig. 5f). Users can control the game character's movements using the keyboard integrated into the clothing, and the keyboards operation remains reliable even when the arm is stretched (as indicated by the red line) or bent (as indicated by the blue line). Additionally, we integrated an interactive display into the system using TSF[tw]. When the game character hits a wall, a red LED is activated as an alert, and when a stage is cleared, a green LED is illuminated (Supplementary Video 4). Next, we demonstrated a large-scale keyboard integrated into the backside of clothing using two strands of TSF[tw] electrodes (Fig. 5g). This unique setup allows the electrical connection of two electrodes upon keyboard tapping, determining the length of the electrical pathway and consequently, the relative resistance (Supplementary Figs. 39 and 40). Utilizing our clothes-integrated keyboard, we successfully typed 'HELLO WORLD.' (Fig. 5g). Finally, we integrated an IMU (inertial measurement unit) into the garment to demonstrate that the TSF[tw] electrodes can reliably transmit signals generated by dynamic motion. By moving the IMU mounted right wrist in various directions, real-time data from each axis was successfully captured without any noticeable drift, latency, jitter, or signal stuttering (Fig. 5h). These demonstrations showcase the practicality, reliability and adaptability of TSF[tw] in constructing diverse types of IDAs customized for specific requirements. This underscores the potential for future wearable clothing advancements.

## Discussion

The large-scale fabrication of stretchable conductive fiber-type electrodes with high conductivity, toughness, and stable conductance is a crucial component for realizing smart digital clothing. In this study, we accomplished our objective by employing a robust method that included: 1) high-power tip sonication-based ink preparation, and 2) a manufacturing technique based on flocculation. This method aimed to fabricate biphasic, tough, and stretchable conducting fibers embedded with LMP. The process involved bridging LMP through the functionalization of TPU via segmentation, inducing phase separation, and encapsulation of LMP. Additionally, the slow evaporation of a high-viscosity solution facilitated the large-scale fabrication of the tough and stretchable conductive fibers. Moreover, the toughness and conductance stability of the fibers could be further enhanced by twisting

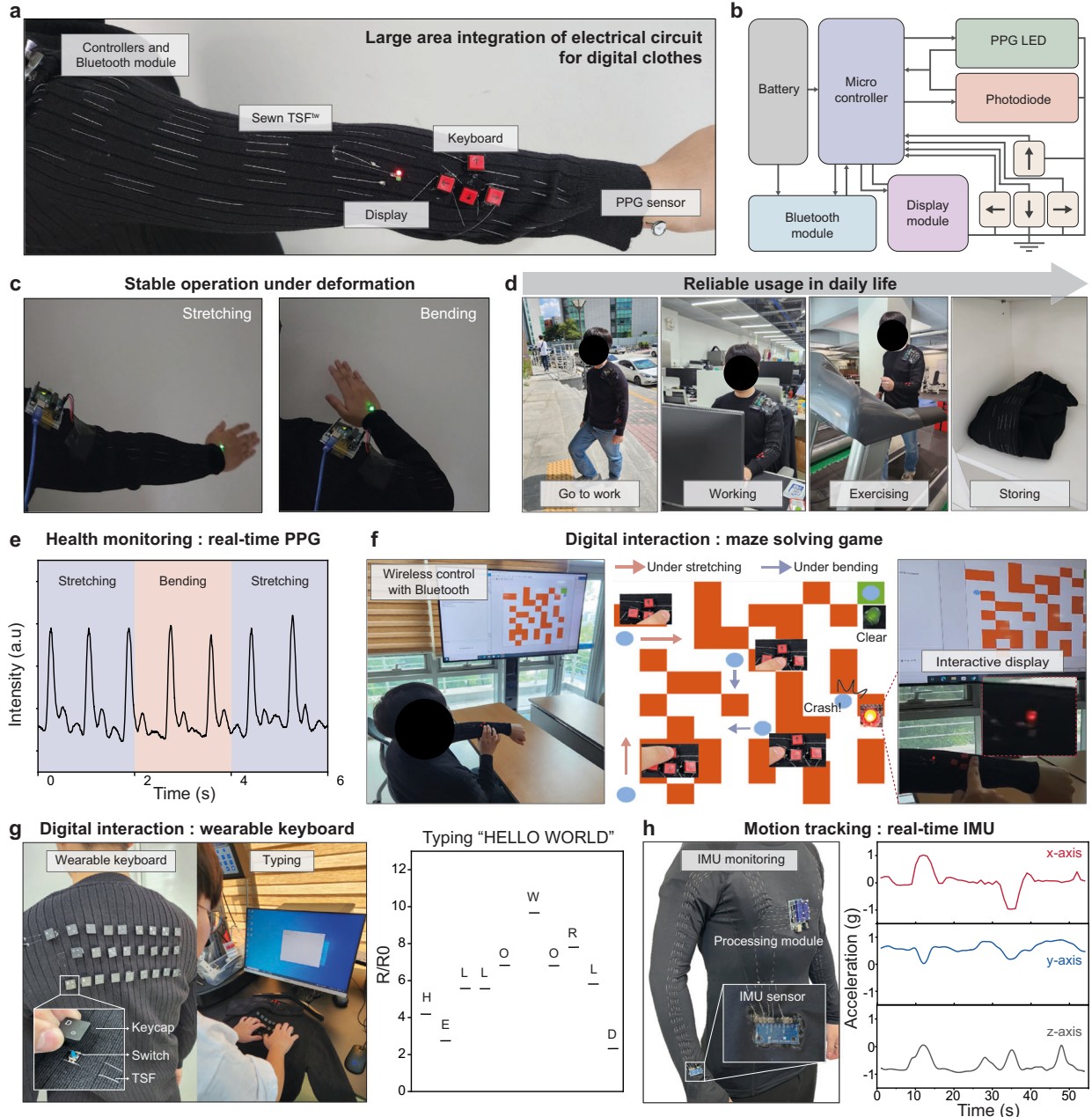

**Fig. 5 | TSF^tw-based intelligent digital apparel (IDA). a** Photograph of twisted TSFs after sintering (TSF^tw)-based IDA. **b** Circuit schematic of IDA. **c** Photograph of IDA operation under mechanical deformation. **d** Photograph of daily life with IDA. **e** Real-time PPG monitoring with IDA under mechanical deformation. **f** Digital interaction system integrated on clothes. **g** Photograph of large-scale wearable keyboard and relative resistance during typing the key "HELLO WORLD". **h** Photograph of motion tracking apparel with IMU, and real-time acceleration recording.

together three plies of TSF, which refers as TSF^tw. Demonstrations of digital functionality integrated into clothing, achieved through the large-scale integration of deformable fiber electrodes, showcase the validity and reliability of TSF^tw, underscoring its potential as a fundamental building block for the development of future smart wearable apparel.

## Methods
### Materials
Unless otherwise stated, all the chemicals used in the current work were used without further purification. Eutectic gallium indium (EGaIn) was purchased from Changsha Ruichi Nonferrous Metals, and Ag flakes (10 μm) were obtained from Alfa Aesar. Thermoplastic polyurethane (TPU, Elastollan® L1185A) was purchased from Goodfellow. SEBS pellets (G1657 M) were purchased from KRATON. Tetrahydrofuran (THF) was purchased from Daejung Chemicals & Metals. N-Methylpyrrolidone (NMP) (99%) was purchased from Sigma-Aldrich.

### Preparation of activated TPU-LMP ink
The activated TPU solution was prepared by dissolving 0.6 g of TPU in 9 mL of NMP at a temperature of 60 °C. This was followed by probe sonication (VC-505, Sonics & Materials, 13 mm microtip) operating at a 70% amplitude for 20 min. In a separate step, 2.4 g of EGaIn was sonicated in 10 mL of acetone at 20% amplitude for 5 min. Once all the acetone had evaporated, the EGaIn microparticles were added and mixed into the NMP- activated TPU solution for 1 min using a Thinky mixer (AR-100).

### Preparation of bare polymer-LMP ink

2.4 g of EGaIn underwent sonication in an acetone solution (10 mL). Upon complete evaporation of the acetone, the resulting EGaIn microparticles were added and thoroughly mixed into the polymer solution comprising bare TPU, SEBS, and Ecoflex.

### Fabrication of free-standing TSF

The fabrication of TSF included a nozzle printing step. After injecting 5 mL of activated TPU-LMP ink into the syringe, nozzle printing was performed on a printing bed featuring a 200 μm microgroove (nozzle diameter: 200 μm, printing speed: 0.5 mm/s). The printing pattern matched the patterned microgroove on the printing bed. Subsequent to the nozzle printing process, the printing bed was positioned on an 80 °C hotplate and left overnight to facilitate the evaporation of NMP. To achieve rapid large-scale fabrication, we produced activated TPU-LMP film on a wafer and then transformed it into fiber form using a laser cutter (3-Axis UV Laser Marker, Keyence). Similarly, we created the bare TPU-LMP fiber, serving as a control, using the exact same procedure.

### Fabrication of AgCF

A total of 2.4 g of silver (Ag) flakes were blended with a TPU solution composed of 0.6 g of TPU and 9 mL of NMP using a Thinky mixer (2000 rpm, 1 min). Subsequently, the resulting mixture was subjected to nozzle printing, employing the same methodology as that used for the TSF, and left to evaporate overnight at a temperature of 80 °C.

### Fabrication of TSF$^{tw}$

The production process for TSF$^{tw}$ was accomplished on a custom-built drawing tower. TSF is rotated at the motor's desired rpm (50–250 rpm) to twist each fiber. The twisted fibers pass through cylindrical heating element heated to 100 °C, which is pulled by the capstan at a speed of 0.5 m/min. This step serves to solidify the twisted arrangement by partially sintering the fibers.

### Encapsulation of TSF$^{tw}$

To prevent unwanted electrical shorting and potential leakage under mechanical stimuli, we encapsulated our fiber with SEBS. To prepare the coating, we made a SEBS solution by dispersing it in cyclohexane. This combination was chosen because cyclohexane effectively dissolves SEBS without interfering with the TPU used for TSF. Additionally, the high volatility of the SEBS/cyclohexane solution facilitates a straightforward and efficient encapsulation process.

### Fabrication of smart digital clothes

To achieve the functionalities of PPG pulse sensing, interactive display, keyboard input, and IMU sensing for digital clothing, a commercially available LED, a PPG module (SEN-11574. Sparkfun), SPST switch button (PB13), and IMU sensor (MPU9250, TDK InvenSense) were incorporated. All communication and control process were managed through Arduino Uno microcontroller interfaced with a Bluetooth module (HC-06, Olimex Ltd).

The connection between the chip and TSF$^{tw}$ was established through the silver epoxy (CW2400, Chemtronics), while a thin layer of epoxy (5-min epoxy, DEVCON) was applied for complete fixation. Each chip was placed in its designated location on a commercially available garment, and the TSF$^{tw}$ was firmly affixed to the clothes through sewing. The detailed circuit information is described in the Supplementary Information.

### Characterization

To analyze the morphology of the TSF and other control fibers, SEM images of the top, bottom, and cross section of the fibers using a Magellan 400 scanning electron microscope (FEI company). For the examination of linking of polymer and LMP, TEM images of activated LMP-TPU solution were obtained using a transmission electron microscope Tecnai F20 (FEI company). Prior to TEM imaging, the LMP-TPU solution was diluted at a volume ratio of 10:1 with NMP.

Chemical characterization was performed using a liquid-state 600 MHz nuclear magnetic resonance (NMR) spectrometer (Avance Neo 600, Bruker BioSpin). 1H NMR analysis was utilized to identify the broken chemical bonds within the TPU.

For electrical characterization, the resistance was measured using a source meter unit (2450, Keithley). Using the voltage bias mode, a constant voltage of 0.1 V was applied and the current was measured with a current limit of 0.5 mA. To measure the conductivity of the LMP-rich conductive path, a fiber sample width of 300 μm was used, and the height of the LMP region (approximately 25 μm) was determined using an SEM image.

A universal testing machine (LS1, Ametek) was employed to precisely apply the desired strain levels and the mechanical characteristics. A 10 N load cell was utilized to measure the force and stretching and releasing rate was fixed at 200% min$^{-1}$ for each measurement. The diameters of fibers were measured through optical microscope, in conjunction with camera software (TCapture) for normalizing the measured force. The toughness of each fiber was measured by calculating the area under the stress-strain curve.

Cycling test was performed using a universal testing machine, applying 0–200% strain at a rate of 20% s$^{-1}$ for 10,000 cycles on 3 cm-long fibers. The electrical resistance was measured by a sourcemter using the voltage bias mode (0.1 V was applied) at a sampling rate of 1 Hz.

### Simulation

To explore the deformation of liquid metals and the alteration in their contact area when subjected to external deformation in a liquid metal composite material, numerical simulations were utilized. The simulations and analytical procedure were performed with MATLAB 2021b, developed by MathWorks.

### Experiments on human subjects

All experiments on wearable applications were performed under approval from the Institutional Review Board at Korea Advanced Institute of Science and Technology (protocol number: KH2021-039).

## Data availability

The authors declare that the data supporting the findings of this study are available within the article and its Supplementary Information files. Source data are provided with this paper.

## Code availability

The authors declare that the custom code is available within the Supplementary Information file and also from the corresponding author upon the request.

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

## Acknowledgements

This work was supported by Samsung Research Funding & Incubation Center of Samsung Electronics (SRFC-IT2401-03 to Seongjun Park.).

## Author contributions

G.-H.L., Y.L., H.S. and S.P. conceived the concept and designed the experiments. G.-H.L., Y.L. and H.S. designed the ink, conducted the experimental work, performed chemical, electrical, and mechanical characterization, and analyzed the data. K.J. developed the code for the digital control system. J.Y. conducted the simulation. S.K. and J.-Y.B. contributed to the experimental work. C.K., C.M., J.K., S.-K.K. and S.R. provided feedback on the manuscript and data analysis. S.P. managed all aspects of the project. G.-H.L., Y.L. and H.S. wrote the draft manuscript. S.P. revised the manuscript. All authors discussed the results and the manuscript.

## Competing interests

The authors declare no competing interests.

## Additional information

[1]Medical Research Center, Seoul National University, Seoul, Republic of Korea. [2]Departments of Cogno-Mechatronics Engineering and Optics & Mechatronics Engineering, Pusan National University, Busan, Republic of Korea. [3]Department of Bio and Brain Engineering, Korea Advanced Institute of Science and Technology (KAIST), Daejeon, Republic of Korea. [4]Program of Brain and Cognitive Engineering, Korea Advanced Institute of Science and Technology (KAIST), Daejeon, Republic of Korea. [5]Department of Mechanical Engineering, Korea Advanced Institute of Science and Technology (KAIST), Daejeon, Republic of Korea. [6]Graduate School of Semiconductor Technology, Korea Advanced Institute of Science and Technology (KAIST), Daejeon, Republic of Korea. [7]Department of Materials Science and Engineering, Seoul National University, Seoul, Republic of Korea. [8]Department of Mechanical Engineering, Carnegie Mellon University, Pittsburgh, PA, USA. [9]Department of Chemistry, Seoul National University, Seoul, Seoul, Republic of Korea. [10]School of Transdisciplinary Innovations, Seoul National University, Seoul, Republic of Korea. [11]Department of Biomedical Science, College of Medicine, Seoul National University, Seoul, Republic of Korea. [12]Interdisciplinary Program in Bioengineering, College of Engineering, Seoul National University, Seoul, Republic of Korea. [13]Department of Transdisciplinary Medicine, Seoul National University Hospital, Seoul, Republic of Korea. [14]These authors contributed equally: Gun-Hee Lee, Yunheum Lee, Hyeonyeob Seo. ✉e-mail: seongjunpark@snu.ac.kr

