## [Transparent Peer Review file · Nature Communications]

Meter-Scale Heterostructure Printing for High-Toughness Fiber Electrodes in Intelligent Digital Apparel

Corresponding Author: Professor Seongjun Park

Version 0:

Reviewer comments:

Reviewer #1

(Remarks to the Author)

In this manuscript, Lee et al. report an approach to fabricate large-scale, high-toughness stretchable and conductive fibers, which consist of the composites of thermoplastic polyurethane (TPU) and liquid metal particles (LMPs). Developing fabrication approaches for high-performance stretchable conductive fibers is important to achieve intelligent digital apparel.

Despite the importance of this work due to the high performance of the fabricated conductive fibers claimed in the manuscript, there are still some concerns to be addressed before this manuscript could be further considered for publication:

1. Although the fabricated fibers have excellent properties, the fiber formation process using printing is not clearly described in the main text, which could make it hard for readers to understand how the fibers are fabricated. There is rich information in the schematic of Figure 1a and corresponding text is suggested to be added to describe the fiber formation process using printing for better understanding.
2. The strong adhesion between the activated TPU and LMPs is the key to preventing LM from leakage. In the manuscript, please explain what kind of key intermolecular interactions cause the strong affinity between activated TPU and gallium oxide when compared with bare TPU, in which polar groups could also interact with LMPs for adhesion.
3. The authors claim no leakage of LM from the fabricated conductive fibers. However, there is a lack of direct evidence for that. Taking SEM images of the scotch tape after the tape test to see if there is liquid metal residue on the tape would be a good way to confirm that.
4. In the manuscript, the authors think phase separation is important to this work. Please comment on why activated TPU can cause phase separation while using bare TPU does not undergo phase separation.
5. Please label the TPU and LMPs in Figure 1c to show both sides of the fibers are enveloped with TPU. The reviewer has difficulty in seeing this point from Figure 1c.
6. In the manuscript, the authors think the LMPs are bridged with activated TPU. If in this case, the insulated TPU materials and oxides on the surface LMPs could block the conductive pathway. Please explain in detail the mechanism that accounts for the conductivity of the fibers.
7. The thickness or diameter of the fibers is an important parameter for fibers. Please comment on what the thickness or diameter of the fabricated fibers are and how small/thin they can be.
8. Please explain how the toughness is measured in the experimental section.
9. In line 322, please double check the spelling of "flaw" or is it meant "flow"?

Reviewer #2

(Remarks to the Author)

I very much enjoyed reading the manuscript, in which the authors have presented their work on conductive and tough fibers for use in digital apparel. The methodology involves blending GaIn liquid metal with TPU and extruding into fibers. These fibers seem to have exceptional performance - stretchable, tough, and highly electrically conductive with very low gauge factor. The key aspects in achieving this seem to be in the partial flocculation of the liquid metal droplets, enabled by their formulation and processing. This partial flocculation reportedly leads to minimal leakage of the liquid metal, high electrical conductivity, and enabling easy electrical connection to be made. The methods and conclusions of the authors appear to be valid.

My recommendation is to publish after some minor corrections. I have included suggestions below that I hope will further improve the work.

In the abstract, lines 38 and 39 could be more accurately written: Our approach involves encapsulating deformable liquid metal particles (LMP) within functionalized thermoplastic polyurethane (TPU).

Throughout the paper, I have wondered how the conductive fibres presented could have both high conductivity without leakage of the LM. The authors mention that 'reliable electrical connection' could be made to the fibres and so on. However, the LM must remain encapsulated within the TPU matrix to prevent leakage. In this case, how can electrical connection be made? Could the authors clarify whether the TPU encapsulation must be penetrated to make electrical connection. Could the authors also comment on whether the conductive fibers must be 'activated' using i.e. strain to induce the conductivity/percolation between encapsulated LMPs. Perhaps a video could also be included showing electrical connection being made/removed at different points along the fibre.

Explanation of mechanical toughness/cracking in conductive fibres with solid conductive fillers (AgCF etc.) between lines 156 and 166 can be improved in terms of readability and validity. First, cracking in these types of composites is due to elastic modulus mismatch. References such as [[https://doi.org/10.1016/0956-7151\(92\)90429-I](https://doi.org/10.1016/0956-7151(92)90429-I)] explains crack formation and propagation in brittle films on compliant substrates. I think the authors could include a similar reference here, outlining crack propagation in composites with compliant matrix and stiff/'brittle' filler. Second, the paragraph is quite disjointed. The points of discussion in the paragraph are as follows. Cracking 'notches' on lines 156 to 159, heterostructure fibers on lines 159 to 161, back to crack formation on 161 to 165, and 'heterostructure TSF' again on 165 to 166.

The authors give conductivity values of $\sim 10^6$. I would like to see the authors' full methodologies on how they measure and calculate this. Aside from the electrical characterisation described (voltage sourced and current measured), what cross sectional area was used in the calculation, and how was the area measured?

Line 173: 'SEBS' was never defined.

Lines 313 and 316: For the sonication in acetone steps, no sonication power or duration were provided.

Line 322: I am unsure of the meaning of the following statement 'on a printing bed characterized by a 200 μ m flaw'. Please could the authors improve the clarity of this?

Reviewer #3

(Remarks to the Author)

Lee et al. developed a wearable conductive fiber with excellent mechanical and electrical properties based on the heterojunction structure formed by bridging TPU with LMP, and achieved applications in various intelligent scenarios, which is a complete and practically meaningful work. However, the manuscript is deemed unsuitable for publication in Nature Communications due to some issues that needed to be addressed and supplemented, as well as some unverifiable data.

I found that the data on strain resistance cycling provided in this work (Supplementary Figure 22) is highly similar to the data from a previous work published by the same authors (Lee GH, et al. Nature Communications, 2023, 14, 4173. DOI: 10.1038/s41467-023-39928-x, Figure 3c). Even though these two sets of data have different horizontal and vertical coordinates, through comparison, I found that they have a significant overlap trend in appearance. As is well known, resistance cycling signals are random and cannot be repeated, even in repeated tests of a same material. In these two works, two devices were prepared under different strategies, but there are overlapping resistance cycling details and trends under different coordinate parameters, which is absolutely impossible. Therefore, I believe that these two works are suspected of data manipulation/falsification. The relevant evidence has been provided in the supplementary documents submitted by us. It should be noted that, due to the lack of raw data and limited image resolution, the above conclusions are all based on visual comparison. Therefore, it is still necessary for editors and authors to verify.

Other issues that need to be addressed:

1. The authors mention "leakage-free" several times in the manuscript (such as lines 76 and 108), while in the application section, the authors explain "To further prevent unwanted electrical shortening or possible leakage...". Please consider whether there is any contradiction or absolutization in this statement.
2. The authors show in the manuscript black fibers (as shown in Figures 1c, 2a, 4b), while the mass-produced fibers exhibit a silver color similar to that after exposure or leakage of liquid metal (as shown in Figures 1b, 2i, 2k, 5a). I'd like to know if this is attributed to the packaging made to prevent short circuits and leaks mentioned in the later paragraph of the article (see line 247 in the application section). If so, the authors should mention it at the beginning of the manuscript to avoid unnecessary misleading information, and further explain whether the advantages of twisted fibers are still significant compared to "conventional liquid metal injection-based stretchable fibers".
3. Figure 4d and Supplementary Figure 18 should be integrated.
4. The labeling of "body deformation" and "engineering strain" in Figure 4d is misleading; They do not directly contain relationships, and the entire strain interval should be labeled as "engineering strain".
5. The authors should improve the bridging effect between LMP and TPU in Figure 2c from the perspective of chemical/physical interactions.
6. The authors are suggested to supplement a cross-sectional morphology of the fibers after twist and sintering.
7. The bridging effect between LMP through TPU in Figure 2e is significantly different from that in Supplementary Figure 12a, and even close to the state in Supplementary Figure 12b. The authors need to provide an explanation for this.

8. In Figure 2e, it is not clear enough to determine the bridging state of TPU solely based on TEM images. The authors are recommended to supplement EDS mappings for characterization and differentiation at the interface.

9. The authors mentioned that liquid metal particles undergo gravity-driven phase separation by activating TPU induced clusters. Is there a quantitative limit to the viscosity of TPU for this effect? Is only a portion of larger volume aggregates able to overcome viscosity and undergo phase separation?

10. SEBS and Cyclohexane need to be mentioned in the materials section.

11. Will liquid metal particles regain the aggregated state after multiple stretches due to frequent contact and compression, thereby posing a risk of side leakage? Authors should supplement the fiber cross-section images after high-strength strain.

Version 1:

Reviewer comments:

Reviewer #1

(Remarks to the Author)

Most of the comments have been addressed. Regarding Comment #3, it's difficult to determine whether there is liquid metal residue from the photograph of the Scotch tape after the peel test. It is recommended to provide a high-resolution microscopic image of the tape for clarity. Additionally, the caption for Supplementary Figure 29 appears to be mislabeled.

Reviewer #2

(Remarks to the Author)

Thanks to the authors for their rebuttal to my initial review. I am satisfied by the authors comments, explanations and alterations to the manuscript. I am also convinced that the stretchable fibres represent a significant contribution to the field. Therefore, my recommendation is to publish.

Reviewer #3

(Remarks to the Author)

The author has systematically replied to all the messages from the reviewers, and this version can be accepted for publication in Nature Communications.

Response to Reviewers' Comments

Title: Heterostructure Printing for Mass Production of High-Toughness Fiber Electrodes in Intelligent Digital Apparel

Authors: Gun-Hee Lee, Yunheum Lee, Hyeonyeob Seo, Kyunghyun Jo, Jinwook Yeo, Semin Kim, Jae-Young Bae, Carmel Majidi, Jiheong Kang, Seung-Kyun Kang, Seunghwa Ryu, and Seongjun Park

Research Article No.: NCOMMS-24-60362

We thank the Reviewers for the careful consideration of our manuscript and the suggested demonstration, clarifications, and analyses that further strengthen our work. Following is a summary of our revision made according to the Reviewers' comments.

The Reviewer's comments are in **bold** and revised texts are highlighted.

Reviewer #1

In this manuscript, Lee et al. report an approach to fabricate large-scale, high-toughness stretchable and conductive fibers, which consist of the composites of thermoplastic polyurethane (TPU) and liquid metal particles (LMPs). Developing fabrication approaches for high-performance stretchable conductive fibers is important to achieve intelligent digital apparel. Despite the importance of this work due to the high performance of the fabricated conductive fibers claimed in the manuscript, there are still some concerns to be addressed before this manuscript could be further considered for publication:

Answer. We appreciate the Reviewer for the helpful comment regarding our work. We have revised our work according to the Reviewer's comments. Point-by-point responses are attached below:

1. Although the fabricated fibers have excellent properties, the fiber formation process using printing is not clearly described in the main text, which could make it hard for readers to understand how the fibers are fabricated. There is rich information in the schematic of Figure 1a and corresponding text is suggested to be added to describe the fiber formation process using printing for better understanding.

Answer. We appreciate the reviewer's question regarding fiber fabrication. As the reviewer pointed out, we have added additional details about the fabrication process in the Methods section and supplementary information. Here, "flaw" refers to the microgroove, which facilitates ink retention for the fabrication of conductive fibers.

Revised Method section (Page 15, Line 19-21)

The fabrication of TSF included a nozzle printing step. After injecting 5 mL of activated TPU-LMP ink into the syringe, nozzle printing was performed on a printing bed featuring a 200 μ m microgroove. The printing pattern matched the patterned microgroove on the printing bed. Subsequent to the nozzle printing process, the printing bed was positioned on an 80 $^{\circ}$ C hotplate and left overnight to facilitate the evaporation of NMP. To achieve rapid large-scale fabrication, we produced activated TPU-LMP film on a wafer and then transformed it into fiber form using a laser cutter (3-Axis UV Laser Marker, Keyence).

Revised Supplementary information

Supplementary Fig. 3| Fabrication methods of TSF.

To create a stable conductor that combines high toughness, stretchability, and conductivity, TSF was designed so that the liquid metal (LM) and polymer phases remain effectively separated while LM particles are stably incorporated into the polymer matrix. This was

achieved by activating TPU chains, thereby promoting favorable interactions between LM and TPU. Leveraging these interactions, we successfully fabricated large quantities of fibers via printing and laser cutting.

Revised Main text (Page 4, Line 19)

In this study, we present a scalable heterostructure printing method for the mass-producing tough and stretchable conductive fibers (TSF) using LM particle (LMP)-embedded thermoplastic polyurethane (TPU) (**Fig. 1a, Supplementary Fig 3**).

2. The strong adhesion between the activated TPU and LMPs is the key to preventing LM from leakage. In the manuscript, please explain what kind of key intermolecular interactions cause the strong affinity between activated TPU and gallium oxide when compared with bare TPU, in which polar groups could also interact with LMPs for adhesion.

Answer. We appreciate the reviewer's insightful comment regarding the interaction between activated TPU and gallium oxide.

To induce the bridging effect between the LMP and the polymer surface, specific chemical groups that can interact with the gallium oxide layer are essential. This bridging effect is critical for forming a robust interfacial linkage that prevents LM leakage. One such group is the amine group, which can form during TPU polymer chain breakage. As shown in Fig. 2d, we observed the breakage of TPU polymer chains during high-power sonication. Furthermore, ¹H NMR analysis confirmed the presence of amine groups. These amine groups play a key role by forming coordination bonds with gallium ions in the oxide layer or hydrogen bonds with surface hydroxyl groups, effectively bridging the interface between activated TPU and LMPs.

In response to the reviewer's comment, we have included the ¹H NMR data and analysis in the Supplementary Information.

Revised Supplementary Information

Supplementary Fig. 9 | Proton nuclear magnetic resonance (¹H NMR) spectrum of TPU before (black) and after tip sonication (red).

Here, the triplet corresponds to the hydrogen peak from the carbon adjacent to the amine group, while the singlet represents the hydrogen peak in the amine group.

Revised Main text (Page 6, Line 11-15)

The profile can be deduced that the functional groups are formed at the end of the TPU activated by sonication, which have a strong affinity for the gallium oxide present on the surface of the LMPs.^{25,26} The proton nuclear magnetic resonance (¹H NMR) spectrum presented in **Supplementary Fig. 9** implies the occurrence of amine group formation in TPU activated through sonication. The H signals, a triplet at 1.22-1.26 p.p.m. and a singlet at 1.67 p.p.m. adjacent to the amine group, serve as evidence of peptide bond cleavage and simultaneous amine group formation.

3. The authors claim no leakage of LM from the fabricated conductive fibers. However, there is a lack of direct evidence for that. Taking SEM images of the scotch tape after the tape test to see if there is liquid metal residue on the tape would be a good way to confirm that.

Answer. We sincerely appreciate the reviewer's suggestion to evaluate the stability of our fiber. In response, we performed the Scotch tape test as recommended. The results confirmed that no residue remained after the peel-off test, and SEM imaging showed no rupture of LMP. We have included these data in the Supplementary Information.

Revised Supplementary Information

Supplementary Fig. 29 | Leakage test by peeling off with Scotch tape. **a.** Photograph of fiber after peel-off test with Scotch tape. **b.** SEM image of the fiber after peeling off with Scotch tape.

Due to the enhanced mechanical stability, there is no rupture of LMP after peeling off with Scotch tape, resulting in no notable residue of LM on the Scotch tape.

Revised Main text (Page 11, Line 8-10)

Finally, we have confirmed the mechanical durability of TSF^{tw} through repeated peel tests with Scotch tape (**Supplementary Fig. 29**) and washing with commercial machine, demonstrating that our TSF^{tw} maintains stable electrical properties even when subjected to external stimuli, which facilitates the realization of practical digital wearable systems (**Fig. 4h**).⁵⁴

4. In the manuscript, the authors think phase separation is important to this work. Please comment on why activated TPU can cause phase separation while using bare TPU does not undergo phase separation.

Answer. We thank the reviewer's comment regarding phase separation. For phase separation to occur, LMP should segregate. However, for reliable printing of highly tough fibers, the ink must have high viscosity; otherwise, it will spread out easily, as shown in Supplementary Fig. 9. With high-viscosity ink, sedimentation of LMP is not easily achieved. To address this, we bridged LMPs with activated TPU, allowing for flocculation, a well-known sedimentation mechanism for particles in solution. The activation of TPU by tip sonication generates amine terminal groups, which attach to the surface of liquid metal particles and form bridges between LMPs. Without this bridging, phase separation of LMP in the TPU would not occur, as demonstrated in Supplementary Fig. 10.

To confirm the bridging of LMP with functionalized TPU, we have included a TEM image showing the bridged LMP.

Revised Supplementary Information

Supplementary Fig. 14| TEM image of LMP. a, LMPs with bridging polymer b, LMPs without bridging polymer.

The amine groups in the functionalized TPU attach to the oxide layer on the surface of the LMP, effectively bridging the particles. Without functionalized TPU, the particles are not bridged and remain dispersed.

Revised Main text (Page 7, Line 4)

Here, we founded that LMP covered with activated TPU, phase separation in the high-viscosity solution is achieved. The bridging between particles establishes physical connections and forms larger clusters (**Supplementary Fig. 14**), resulting in enforced phase separation, referred to as flocculation (**Fig. 2f**).^{28,29} **The flocculation is driven by the**

bridging effect of activated TPU, in which terminal amine groups on TPU chains form interparticle linkages between adjacent LMPs, effectively inducing LMP clusters that promotes phase separation even in a high-viscosity solution.

5. Please label the TPU and LMPs in Figure 1c to show both sides of the fibers are enveloped with TPU. The reviewer has difficulty in seeing this point from Figure 1c.

Answer. We thank the reviewer's comment on enhancing visibility. We have revised Figure 1c by adding labels to improve the visibility of both sides of the fibers.

6. In the manuscript, the authors think the LMPs are bridged with activated TPU. If in this case, the insulated TPU materials and oxides on the surface LMPs could block the conductive pathway. Please explain in detail the mechanism that accounts for the conductivity of the fibers.

Answer. We appreciate the reviewer's comment regarding the conductivity of our fiber. As the reviewer mentioned, since the LMP is encapsulated by TPU, the fiber initially does not exhibit conductivity. However, due to the dense packing of LMP at the bottom side of the fiber, a conductive pathway can be easily established through particle sedimentation when strain is applied. Upon applying strain, the particles elongate, the encapsulating polymer also stretches, and the surface oxide layer is ruptured. Since EGaIn has a very high surface tension, the elongated particles connect to each other through the ruptured oxide layer, forming a conductive pathway similar to conventional methods [Adv. Funct. Mater. 2309660 (2023), Nat. Commun, 1300 (2019)].

We have included a side SEM image of our fiber to illustrate the compact assembly of LMP, which forms a conductive pathway.

Revised Supplementary Information

Supplementary Fig. 15| Side-view SEM images of TSF.

The side SEM images of TSF display the dense arrangement of LMP, which creates a conductive pathway within the TPU.

Revised Main text (Page 7, Line 8)

The separated Fig. 2h and Supplementary Fig. 15 present a cross-sectional SEM image of the hetero-structured TSF. The SEM image visualizes the densely packed LMP region at the bottom, which facilitates the formation of a conductive pathway through strain-induced elongation and connection of the particles.

7. The thickness or diameter of the fibers is an important parameter for fibers. Please comment on what the thickness or diameter of the fabricated fibers are and how small/thin they can be.

Answer. We appreciate the reviewer's comment regarding the conductivity of our fiber. As the reviewer mentioned, the thickness and diameter of the fiber are important factors for its practical use. If the fiber is too thin, it can easily tear or break under minimal mechanical force. By adjusting the solvent-to-polymer ratio, the viscosity of the ink can be tuned. Increasing the viscosity allows for the fabrication of thicker fibers. In this experiment, we used a medium-viscosity ink that achieved approximately 150 μm thickness, as this viscosity works well for both the printing and film formation steps required for laser cutting. To demonstrate the tunability of the fabrication process, we have added side-view images of fibers with different viscosities, showing the effect of viscosity on the fiber structure. This data is included in the Supplementary Information.

Revised Supplementary Information

Supplementary Fig. 16 Thickness control of fiber.

By controlling the viscosity of the solution, the thickness of the fiber can be adjusted.

Revised Main text (Page 7, Line 9-10)

Furthermore, this phenomenon occurs across a range of solution viscosities, enabling precise tuning of fiber thickness (**Supplementary Fig. 16**).

8. Please explain how the toughness is measured in the experimental section.

Answer. We thank the reviewer for the comment regarding the measurement of toughness. In this study, we monitor the stress-strain curve of the fiber using a universal testing machine (LS1, Ametek). After obtaining the stress-strain curve, we calculate the toughness by determining the area under the curve. We have included this information in the Supplementary Information.

Revised Method section (Page 18, Line 1-4)

A universal testing machine (LS1, Ametek) was employed to precisely apply the desired strain levels and the mechanical characteristics. A 10N load cell was utilized to measure the force and stretching and releasing rate was fixed at $200\% \text{ min}^{-1}$ for each measurement. The diameters of fibers were measured through optical microscope, in conjunction with camera software (TCapture) for normalizing the measured force. The toughness of each fiber was measured by calculating the area under the stress-strain curve.

9. In line 322, please double check the spelling of “flaw” or is it meant “flow”?

Answer. We thank the reviewer for the comment regarding the clarity of the wording. Here, "flaw" refers to the microgroove in the printing bed. To further enhance the clarity of our work, we have revised the Method Section.

Revised Method section (Page 15, Line 19-21)

The fabrication of TSF included a nozzle printing step. After injecting 5 mL of activated TPU-LMP ink into the syringe, nozzle printing was performed on a printing bed featuring a $200\mu\text{m}$ microgroove. The printing pattern matched the patterned microgroove on the printing bed

Reviewer #2

I very much enjoyed reading the manuscript, in which the authors have presented their work on conductive and tough fibers for use in digital apparel. The methodology involves blending GaIn liquid metal with TPU and extruding into fibers. These fibres seem to have exceptional performance - stretchable, tough, and highly electrically conductive with very low gauge factor. The key aspects in achieving this seem to be in the partial flocculation of the liquid metal droplets, enabled by their formulation and processing. This partial flocculation reportedly leads to minimal leakage of the liquid metal, high electrical conductivity, and enabling easy electrical connection to be made. The methods and conclusions of the authors appear to be valid.

My recommendation is to publish after some minor corrections. I have included suggestions below that I hope will further improve the work.

Answer. We appreciate the Reviewer's insightful feedback on our work. We have addressed the comments thoroughly and incorporated the necessary revisions. Detailed point-by-point responses are provided below:

1. In the abstract, lines 38 and 39 could be more accurately written: Our approach involves encapsulating deformable liquid metal particles (LMP) within functionalized thermoplastic polyurethane (TPU).

Answer. We thank the reviewer's comment about abstract. Here, we have revised abstract as reviewer suggested.

Revised Abstract (Page 3, Line 7-8)

In this study, we present a novel heterostructure printing method capable of mass-producing (~50m) biphasic conductive fibers meeting these exacting criteria. **Our approach involves encapsulating deformable liquid metal particles (LMP) within functionalized thermoplastic polyurethane (TPU).**

2. Throughout the paper, I have wondered how the conductive fibres presented could have both high conductivity without leakage of the LM. The authors mention that ‘reliable electrical connection’ could be made to the fibres and so on. However, the LM must remain encapsulated within the TPU matrix to prevent leakage. In this case, how can electrical connection be made? Could the authors clarify whether the TPU encapsulation must be penetrated to make electrical connection.

Answer. We appreciate the reviewer’s comment regarding conductivity and stability. As the reviewer mentioned, since the LMP is encapsulated by TPU, the fiber initially does not exhibit conductivity. However, due to the dense packing of LMP at the bottom of the fiber, a conductive pathway can be established through particle sedimentation when strain is applied. When strain is applied, the particles elongate, causing the encapsulating polymer to stretch and the surface oxide layer to rupture. Because EGaIn has very high surface tension, the elongated particles connect to each other through the ruptured oxide layer, forming a conductive pathway similar to conventional methods [Adv. Funct. Mater. 2309660 (2023), Nat. Commun, 1300 (2019)]. We have included a side SEM image of our fiber to demonstrate the formation of this conductive pathway. Additionally, since the bottom sides of the particles are densely encapsulated by the polymer, the fiber exhibits enhanced mechanical stability. To further demonstrate this, we have conducted a peel-off test, which shows the improved stability of our fiber.

For electrical conductivity, even though all the LMP is encapsulated, the very bottom side of the LMP remains slightly exposed, allowing electrical contact with conventional electronic chips. To illustrate this, we have included side and bottom SEM images of our fiber in the Supplementary Information.

Revised Supplementary Information

Supplementary Fig. 15| Side-view SEM images of TSF.

The side SEM images of TSF display the dense arrangement of LMP, which creates a conductive pathway within the TPU.

Revised Main text (Page 7, Line 8)

The separated **Fig. 2h** and **Supplementary Fig. 15** present a cross-sectional SEM image of the hetero-structured TSF.

3. Could the authors also comment on whether the conductive fibers must be 'activated' using i.e. strain to induce the conductivity/percolation between encapsulated LMPs. Perhaps a video could also be included showing electrical connection being made/removed at different points along the fibre.

Answer. We appreciate the reviewer's comment on the electrical activation of our fiber. Since our fiber is initially encapsulated by TPU and an oxide layer, it requires electrical activation using stimuli, similar to conventional LMP. However, as the fiber must be detached from the printing bed, this detachment process itself is sufficient to electrically activate the fiber, offering a more practical approach compared to electrical circuit activation. We have included *a video and image* in the supplementary information to illustrate the electrical activation process.

Revised Main text (Page 7, Line 13-15)

During fiber peel-off for free-standing applications, the LMP is electrically connected to the localized strain (**Supplementary Fig. 17, Supplementary Video 2**).

Revised Supplementary Information

Supplementary Fig. 17 | Photograph of the electrical connection of the TSF during the peel-off process.

4. Explanation of mechanical toughness/cracking in conductive fibres with solid conductive fillers (AgCF etc.) between lines 156 and 166 can be improved in terms of readability and validity. First, cracking in these types of composites is due to elastic modulus mismatch. References such as [[https://doi.org/10.1016/0956-7151\(92\)90429-I](https://doi.org/10.1016/0956-7151(92)90429-I)] explains crack formation and propagation in brittle films on compliant substrates. I think the authors could include a similar reference here, outlining crack propagation in composites with compliant matrix and stiff/‘brittle’ filler. Second, the paragraph is quite disjointed. The points of discussion in the paragraph are as follows. Cracking ‘notches’ on lines 156 to 159, heterostructure fibers on lines 159 to 161, back to crack formation on 161 to 165, and ‘heterostructure TSF’ again on 165 to 166.

Answer. We appreciate the reviewer's valuable feedback. As suggested, we have revised the manuscript to enhance its logical flow and clarity. Additionally, we have incorporated the recommended references.

Revised Main Text (Page 8, Line 4-13)

This behavior contrasts with that of TSF, which demonstrates superior stretchability and toughness (**Fig. 3b**). We attribute this difference in mechanical performance to the presence or absence of a stiff filler. In fibers incorporating a stiff solid filler, the mismatch in elastic modulus induces stress concentration, facilitating crack initiation and propagation, thereby compromising mechanical toughness.³⁵ In contrast, in TSF, all additives exhibit deformability comparable to that of the polymer matrix, preventing the formation of voids or microholes that could act as stress concentrators within the elastomer structure. Furthermore, the conductive LMP-rich region is spatially segregated from the tough polymer matrix, as illustrated in the subsequent figure. Consequently, TSF retains mechanical properties comparable to those of pure TPU fiber, ensuring both structural integrity and functional performance.

Added Reference

35 Thouless, M. D., Olsson, E. & Gupta, A. CRACKING OF BRITTLE FILMS ON ELASTIC SUBSTRATES. *Acta Metallurgica Et Materialia* **40**, 1287-1292, doi:10.1016/0956-7151(92)90429-i (1992).

5. The authors give conductivity values of $\sim 10^6$. I Would like to see the authors' full methodologies on how they measure and calculate this. Aside from the electrical characterisation described (voltage sourced and current measured), what cross sectional area was used in the calculation, and how was the area measured?

Answer. We appreciate the reviewer's question regarding our methodology. To measure electrical conductivity, we calculated the cross-sectional area. For accuracy, we used an SEM image and considered only the LMP region to precisely determine the conductive area. The width corresponds to the TSF's width of 300 μm , while the height is approximately 25 μm , including only the LMP region and excluding the TPU-rich region. We have revised the Methods section to incorporate this additional information.

Revised Method section (Page 17, Line 19-21)

For electrical characterization, the resistance was measured using a source meter unit (2450, Keithley). Using the voltage bias mode, a constant voltage of 0.1V was applied and the current was measured with a current limit of 0.5mA. To measure the conductivity of the LMP-rich conductive path, a fiber sample width of 300 μm was used, and the height of the LMP region (approximately 25 μm) was determined using an SEM image.

6. 'SEBS' was never defined.

Answer. We thank the reviewer for the careful review. We have revised the main text to include the full term of SEBS (Styrene-Ethylene-Butylene-Styrene).

Revised Main text (Page 8, Line 20)

Other fibers containing LMP and elastomers such as Styrene-Ethylene-Butylene-Styrene (SEBS) or ecoflex also exhibit lower toughness, making them less suitable for practical use due to both their insufficient toughness and difficulties in the large-scale fabrication process.

7. Lines 313 and 316: For the sonication in acetone steps, no sonication power or duration were provided.

Answer. We appreciate your valuable feedback regarding fabrication method. We have revised the method section.

Revised Method section (Page 15, Line 10-11)

Preparation of activated TPU-LMP ink. The activated TPU solution was prepared by dissolving 0.6g of TPU in 9mL of NMP at 60°C. This mixture was then subjected to probe sonication (VC-505, Sonics & Materials, 13 mm microtip) at 70% amplitude for 20 minutes. **In a separate step, 2.4g of EGaIn was sonicated in 10mL of acetone at 20% amplitude for 5 minutes.** After the complete evaporation of the acetone, the EGaIn microparticles were added to the NMP-activated TPU solution and mixed for 1 minute using a Thinky mixer (AR-100).

8. Line 322: I am unsure of the meaning of the following statement ‘on a printing bed characterized by a 200µm flaw’. Please could the authors improve the clarity of this?

Answer. We thank the reviewer for the comment regarding the clarity of the wording. Here, "flaw" refers to the microgroove in the printing bed. To further enhance the clarity of our work, we have revised the Method Section.

Revised Method section (Page 15, Line 19-21)

Fabrication of free-standing TSF. The fabrication of TSF included a nozzle printing step. After injecting 5 mL of activated TPU-LMP ink into the syringe, nozzle printing was performed on a printing bed featuring a 200µm microgroove (nozzle diameter: 200µm, printing speed: 0.5 mm/s). The printing pattern matched the patterned microgroove on the printing bed.

Reviewer #3

Lee et al. developed a wearable conductive fiber with excellent mechanical and electrical properties based on the heterojunction structure formed by bridging TPU with LMP, and achieved applications in various intelligent scenarios, which is a complete and practically meaningful work. However, the manuscript is deemed unsuitable for publication in Nature Communications due to some issues that needed to be addressed and supplemented, as well as some unverifiable data.

I found that the data on strain resistance cycling provided in this work (Supplementary Figure 22) is highly similar to the data from a previous work published by the same authors (Lee GH, et al. Nature Communications, 2023, 14, 4173. DOI: 10.1038/s41467-023-39928-x, Figure 3c). Even though these two sets of data have different horizontal and vertical coordinates, through comparison, I found that they have a significant overlap trend in appearance. As is well known, resistance cycling signals are random and cannot be repeated, even in repeated tests of a same material. In these two works, two devices were prepared under different strategies, but there are overlapping resistance cycling details and trends under different coordinate parameters, which is absolutely impossible. Therefore, I believe that these two works are suspected of data manipulation/falsification. The relevant evidence has been provided in the supplementary documents submitted by us. It should be noted that, due to the lack of raw data and limited image resolution, the above conclusions are all based on visual comparison. Therefore, it is still necessary for editors and authors to verify.

Answer. We would like to sincerely thank the reviewer for their thorough and careful evaluation, particularly for identifying the duplication in our dataset. Upon further investigation, we discovered a technical issue in our cycle test setup, which led to the overlap of duplicate data in our Excel file. We deeply regret this oversight and have since rechecked all of our data thoroughly and recollected the dataset.

This was a significant oversight on our part. Due to the large size of the dataset, we initially overlooked the duplication. Moreover, since the affected data appeared in the supplementary information rather than the main figures, we did not give it the attention it deserved. We now fully recognize the importance of this mistake and take complete responsibility for the error.

We are grateful to the reviewer for their keen observation, which enabled us to identify and rectify the issue prior to publication. In response, we have carefully reviewed and replaced all affected data, re-examined our experimental setup, and implemented additional measures to ensure the accuracy and integrity of our results moving forward.

We deeply apologize for this mistake and sincerely appreciate the opportunity to address it

before publication. To prevent any future issues, we have reset our experimental setups, recollected the data to ensure its reliability, and will exercise even greater care in verifying our data. The feedback provided by the reviewer has been invaluable in improving the quality of our research, and we are truly grateful for it.

As part of our corrective actions, we have recollected the cycle data and revised the supplementary information accordingly. Cyclic tests were conducted on 3 cm-long fibers using a universal testing machine, applying 0–200% strain at a rate of $20\% \text{ s}^{-1}$ for 10,000 cycles. The electrical resistance was measured by a sourcemeter at a sampling rate of 1 Hz, and additional details are provided in the Methods section. Although a slight baseline drift of approximately 2% was observed over 10,000 cycles, the hysteresis remained minimal, confirming the conductor’s stability. This behavior can be attributed to the rapid reestablishment of conductive pathways enabled by the liquid-like characteristics of the liquid metal, coupled with the low hysteresis of the twisted structure.

Revised Supplementary Information

Supplementary Fig. 27| Relative resistance of TSFtwS under applications of 200% stretching.

Revised Method section (Page 18, Line 5-7)

Cycling test was performed using a universal testing machine, applying 0-200% strain at a rate of $20\% \text{ s}^{-1}$ for 10,000 cycles on 3 cm-long fibers. The electrical resistance was measured by a sourcemeter using the voltage bias mode (0.1V was applied) at a sampling rate of 1 Hz

1. The authors mention "leakage-free" several times in the manuscript (such as lines 76 and 108), while in the application section, the authors explain "To further prevent unwanted electrical shortening or possible leakage...". Please consider whether there is any contradiction or absolutization in this statement.

Answer. We appreciate the Reviewer's comment on the clarity of our work. In this study, the particles are densely packed with activated TPU, which effectively prevents leakage under mechanical stimuli. To avoid unwanted electrical shorting in the applications section, we implemented additional encapsulation with the polymer. While our fibers demonstrate enhanced mechanical stability, there is still a potential for leakage under extreme shear forces. To mitigate this, we included further encapsulation during the application process.

In response to the Reviewer's concerns, we have revised the sentence for clarity and added supplementary information, demonstrating that there is no residue or particle rupture after peeling off with Scotch tape.

Revised Main Text (Page 11, Line 8-10, Line 23-25)

Finally, we have confirmed the mechanical durability of TSF^{tw} through repeated peel tests with Scotch tape (Supplementary Fig. 29) and washing with commercial machine, demonstrating that our TSF^{tw} maintains stable electrical properties even when subjected to external stimuli, which facilitates the realization of practical digital wearable systems (Fig. 4h).⁵⁴

To further prevent unwanted electrical shorting or potential leakage under extreme mechanical stimuli, we developed and employed a fast, on-the-fly encapsulation method for TSF^{tw} using SEBS. (Supplementary Figs. 33 and 34).

Revised Supplementary Information

Supplementary Fig. 29| Leakage test by peeling off with Scotch tape. a. Photograph of fiber after peel-off test with Scotch tape. b. SEM image of the fiber after peeling off with Scotch tape.

Due to the enhanced mechanical stability, there is no rupture of LMP after peeling off with Scotch tape, resulting in no notable residue of LM on the Scotch tape.

2. The authors show in the manuscript black fibers (as shown in Figures 1c, 2a, 4b), while the mass-produced fibers exhibit a silver color similar to that after exposure or leakage of liquid metal (as shown in Figures 1b, 2i, 2k, 5a). I'd like to know if this is attributed to the packaging made to prevent short circuits and leaks mentioned in the later paragraph of the article (see line 247 in the application section). If so, the authors should mention it at the beginning of the manuscript to avoid unnecessary misleading information, and further explain whether the advantages of twisted fibers are still significant compared to “conventional liquid metal injection-based stretchable fibers”.

Answer. We appreciate the reviewer’s insightful comments. The fiber used in our study remains unchanged across all images, with no leakage of LMP. The observed color variations are solely attributed to differences in background and lighting conditions during photography, rather than any inherent change in the fiber itself. To clarify this, we have included a comparative photograph demonstrating how the fiber’s appearance shifts depending on the surrounding background and lighting conditions.

Moreover, the incorporation of activated TPU as a binder for LMP significantly enhances the mechanical stability of the fiber compared to conventional liquid-state EGaIn injection, which poses challenges in establishing reliable electrical connections with standard electronic components (as shown in Supplementary Fig. 2). Unlike liquid-state EGaIn, which lacks structural integrity and is susceptible to leakage, our fiber demonstrates exceptional mechanical robustness, enabling seamless integration with conventional chips and facilitating the development of functional smart textiles (Fig. 5).

To further validate the improved mechanical resilience of our approach, we conducted a Scotch tape peeling test. The results confirmed that the fiber left no residual material after peeling, underscoring its superior adhesion and mechanical durability in comparison to traditional liquid-state EGaIn-based methods.

Additionally, we have revised and expanded the data on stability in the Supplementary Information to further clarify the significance of our work.

Supplementary Information

Supplementary Fig. 2| Leakage of LM-injected fiber during integration with electronic chip.

The LM-injected fiber is encapsulated with an elastomer, which must be removed when integrating the fiber with an electronic chip featuring a conductive LM layer. However, the removal of this encapsulation layer can result in bulk LM leakage due to its inherent fluidity, thereby compromising the stability and reliability of integration with solid conventional chips.

Revised Supplementary Information

Supplementary Fig. 29| Leakage test by peeling off with Scotch tape. a. Photograph of fiber after peel-off test with Scotch tape. **b.** SEM image of the fiber after peeling off with Scotch tape.

Due to the enhanced mechanical stability, there is no rupture of LMP after peeling off with Scotch tape, resulting in no notable residue of LM on the Scotch tape.

Revised Main text (Page 11, Line 8-10)

Finally, we have confirmed the mechanical durability of TSF^{tw} through repeated peel tests with Scotch tape (Supplementary Fig. 29) and washing with commercial machine,

demonstrating that our TSF^{tw} maintains stable electrical properties even when subjected to external stimuli, which facilitates the realization of practical digital wearable systems (**Fig. 4h**).⁵⁴

3. Figure 4d and Supplementary Figure 18 should be integrated.

Answer. We appreciate the reviewer’s comment regarding the data display. As suggested, we have integrated Fig. 4d with Supplementary Fig. 18 and removed Supplementary Fig. 18

4. The labeling of "body deformation" and "engineering strain" in Figure 4d is misleading; They do not directly contain relationships, and the entire strain interval should be labeled as “engineering strain”.

Answer. We appreciate the reviewer’s comment on enhancing the clarity of our work. In this study, we used the term "engineering strain" to refer to the strain of interest in terms of practical usage, as body deformation typically does not exceed 200%.

As suggested by the reviewer, we have removed the term "engineering strain" and further explained why we characterized most of the electromechanical properties up to 200% in the Main Text.

Revised Main Text (Page 9, Line 8-10)

Our experimental results confirm that a three-ply configuration of TSF is the optimal design, finding that increasing the ply count beyond four does not result in further enhancements in toughness (Fig. 4d). Here, we note that smart clothing applications primarily rely on body deformation, which is typically less than 200%. Therefore, our focus is on the strain region below this threshold.

5. The authors should improve the bridging effect between LMP and TPU in Figure 2c from the perspective of chemical/physical interactions.

Answer. We appreciate the reviewer's insightful comment regarding the interaction between activated TPU and gallium oxide.

Achieving strong affinity between gallium oxide and the polymer is essential for forming a stable interface and promoting a bridging effect between LMPs and TPU.

Achieving strong affinity between gallium oxide and polymer, which is essential for promoting a bridging effect between LMPs and TPU, requires specific chemical groups capable of interacting with the oxide layer. One such group is the amine group, which can form during TPU polymer chain breakage. As shown in Fig. 2d, we observed the breakage of TPU polymer chains during high-power sonication. Furthermore, ¹H NMR analysis confirmed the presence of amine groups. Since amine groups can interact with the oxide layer on the surface of LMPs, activated TPU with terminal amine groups is capable of bridging LMPs through chemical bonding. In particular, the lone pair electrons on the nitrogen atom of the amine group can form coordination bonds with gallium ions in the gallium oxide layer, or engage in hydrogen bonding with surface hydroxyl groups.

In response to the reviewer's comment, we have included the ¹H NMR data and analysis in Supplementary Information.

Revised Supplementary Information

Supplementary Fig. 9 | Proton nuclear magnetic resonance (¹H NMR) spectrum of TPU before (black) and after tip sonication (red).

Here, the triplet corresponds to the hydrogen peak from the carbon adjacent to the amine group, while the singlet represents the hydrogen peak in the amine group.

Revised Main text (Page 6, Line 11-15)

The profile can be deduced that the functional groups are formed at the end of the TPU activated by sonication, which have a strong affinity for the gallium oxide present on the surface of the LMPs.^{25,26} The proton nuclear magnetic resonance (¹H NMR) spectrum presented in **Supplementary Fig. 9** implies the occurrence of amine group formation in TPU activated through sonication. The H signals, a triplet at 1.22-1.26 p.p.m. and a singlet at 1.67 p.p.m. adjacent to the amine group, serve as evidence of peptide bond cleavage and simultaneous amine group formation.

6. The authors are suggested to supplement a cross-sectional morphology of the fibers after twist and sintering.

Answer. We thank the reviewers for their comment regarding the morphology. Capturing the vertical cross-section of the fiber is impractical, so we have included a photo taken in the horizontal direction.

Revised Supplementary Information

a

Supplementary Fig. 33| SEBS coating on TSF^{tw}. a, Cross-sectional OM image for SEBS-coated TSF^{tw}.

7. The bridging effect between LMP through TPU in Figure 2e is significantly different from that in Supplementary Figure 12a, and even close to the state in Supplementary Figure 12b. The authors need to provide an explanation for this.

Answer. We appreciate the Reviewer's comment regarding the bridging effect between LMP through TPU. To address this concern, we have conducted additional TEM imaging to further investigate the bridging phenomenon.

The TEM results indicate that polymer bridging occurs only in the presence of activated TPU subjected to tip sonication. Tip sonication induces polymer chain scission, leading to the generation of terminal amine groups, these amine groups can interact with the gallium oxide layer on the LMP surface. In contrast, when bare TPU is used no such interaction occurs, resulting in poor contact between LMPs and the polymer. This confirms that the bridging effect is not observed in the absence of tip-sonicated TPU, providing a more accurate understanding of the mechanism.

To provide accurate information, we have revised the Supplementary Information section as detailed below:

8. In Figure 2e, it is not clear enough to determine the bridging state of TPU solely based on TEM images. The authors are recommended to supplement EDS mappings for characterization and differentiation at the interface.

Answer. We sincerely appreciate the reviewer's insightful comment and the opportunity to clarify this point.

To further characterize the bridging state of thermoplastic polyurethane (TPU), we performed TEM-EDS analysis on an activated TPU-LMP ink sample prepared under identical conditions, diluted with NMP at a 10:1 volume ratio.

Elemental mapping was conducted for carbon, gallium, and indium to differentiate the TPU and liquid metal particles (LMP) interface. The TEM-EDS results confirmed that the liquid metal particles (LMPs) were primarily mapped to gallium and indium, while carbon signals were observed at the LMP surface. This provides direct evidence that TPU is effectively bridged to the LMP surface.

To enhance clarity and provide additional supporting data, we have revised the Supplementary Information section accordingly, as detailed below:

Revised Main text (Page 6, Line 17)

The activated TPU efficiently encapsulates the LMPs and forms bridges between particles, as evidenced by the TEM image (Fig. 2e) and TEM energy dispersive X-ray spectroscopy (EDS) (Supplementary Fig. 10).

9. The authors mentioned that liquid metal particles undergo gravity-driven phase separation by activating TPU induced clusters. Is there a quantitative limit to the viscosity of TPU for this effect? Is only a portion of larger volume aggregates able to overcome viscosity and undergo phase separation?

Answer. Thank you for your insightful comment and question. As the reviewer mentioned, viscosity is closely related to the sedimentation of LMP. If we increase the viscosity, aggregated particles are less likely to undergo sedimentation.

Stokes' law states that particles of the same size settle more slowly as the viscosity of the medium increases. If the film solidifies before particle settling occurs, phase separation may be suppressed.

$$\text{Stoke's law : } v = \frac{2}{9\mu}(\rho_p - \rho_f)r^2g$$

(where v is the settling velocity, g is gravitational field strength, r is the radius of the spherical particle, ρ_p is the mass density of the particle, ρ_f is the mass density of the fluid, μ is the dynamic viscosity of fluid)

In our study, we increased the effective particle size by inducing the formation of liquid metal particle (LMP) clusters through TPU activation, enabling the formation of large agglomerates that can settle more rapidly, even in highly viscous media. This mechanism can be described as phase separation by flocculation, in which each particle settles under gravity as part of a loosely bound aggregate.

However, excessively increasing the TPU content to enhance viscosity renders solution preparation impractical. High viscosity interferes with high-power sonication, preventing effective TPU activation and leading to gelation rather than forming a stable solution. Additionally, once the solution reaches a gel-like state, it becomes unsuitable for fiber or film formation.

In our study, we determined that increasing viscosity up to the point of LMP formation allowed phase separation to be observed. Specifically, phase separation occurred in solutions with concentrations up to 0.15 g/mL, which we identified as the upper limit for ink fabrication. In response to the reviewer's concerns, we have revised the corresponding sentence for clarity and provided additional data in the Supplementary Information.

Supplementary Fig. 16 | Thickness control of fiber.

By controlling the viscosity of the solution, the thickness of the fiber can be adjusted.

Revised Main text (Page 7, Line 9-10)

Furthermore, this phenomenon occurs across a range of solution viscosities, enabling precise tuning of fiber thickness (**Supplementary Fig. 16**).

10. SEBS and Cyclohexane need to be mentioned in the materials section.

Answer. We appreciate the reviewer's careful review. We have added the information about SEBS and cyclohexane in the method section.

Revised Method section (Page 15, Line 6-7)

Materials. Unless otherwise stated, all the chemicals used in the current work were used without further purification. Eutectic gallium indium (EGaIn) was purchased from Changsha Ruichi Nonferrous Metals, and Ag flakes (10 μm) were obtained from Alfa Aesar. Thermoplastic polyurethane (TPU, Elastollan® L1185A) was purchased from Goodfellow. SEBS pellets (G1657 M) were purchased from KRATON. Tetrahydrofuran (THF) was purchased from Daejung Chemicals & Metals. N-Methylpyrrolidone (NMP) (99%) was purchased from Sigma-Aldrich.

11. Will liquid metal particles regain the aggregated state after multiple stretches due to frequent contact and compression, thereby posing a risk of side leakage? Authors should supplement the fiber cross-section images after high-strength strain.

Answer. We appreciate the reviewer’s insightful comments. As mentioned by the reviewer, we conducted additional experiments to examine leakage at 350% strain. The results confirm that there was no leakage or rupture due to the high-strength strain. We only observed wrinkling on the bottommost thin TPU layer.

Here, we have included supplementary information on leakage, along with SEM images of the samples after high-strain testing (350% stretching at a rate of 350%/min), for further reference.

Revised Supplementary Information

Supplementary Fig. 21| SEM image and photograph of TSF after 350% stretching. a, b SEM image of TSF after stretching. c, Photograph of TSF after stretching

SEM image and photograph of TSF after 350% stretching confirm that, even at 350% strain, the polymer encapsulation on the bottom surface allows the structure to remain intact without leakage of the LMP.

Revised Main text (Page 9, Line 13-15)

Furthermore, the activated TPU encapsulates the LMP, effectively preventing particle leakage or rupture under high-strain conditions (**Supplementary Fig. 21**)

Response to Reviewers' Comments

Title: Heterostructure Printing for Mass Production of High-Toughness Fiber Electrodes in Intelligent Digital Apparel

Authors: Gun-Hee Lee, Yunheum Lee, Hyeonyeob Seo, Kyunghyun Jo, Jinwook Yeo, Semin Kim, Jae-Young Bae, Chul Kim, Carmel Majidi, Jiheong Kang, Seung-Kyun Kang, Seunghwa Ryu, and Seongjun Park

Research Article No.: NCOMMS-24-60362

We thank the Reviewers for the careful consideration of our manuscript and the suggested demonstration, clarifications, and analyses that further strengthen our work. Following is a summary of our revision made according to the Reviewers' comments.

The Reviewer's comments are in **bold** and revised texts are **highlighted**.

Reviewer #1

Most of the comments have been addressed. Regarding Comment #3, it's difficult to determine whether there is liquid metal residue from the photograph of the Scotch tape after the peel test. It is recommended to provide a high-resolution microscopic image of the tape for clarity. Additionally, the caption for Supplementary Figure 29 appears to be mislabeled.

Answer. We sincerely thank the reviewer for their detailed and constructive feedback. We have added an optical image of scotch tape to supplementary figure 29 in response to your comment and corrected the mislabeled caption. The following is the revised supplementary information

Revised Supplementary Information

Supplementary Fig. 29| Leakage test by peeling off with Scotch tape. a. SEM image of the fiber after peeling off with Scotch tape. b. Photograph of fiber after peel-off test with Scotch tape. c. Optical image of scotch tape after peeling off fiber.

Reviewer #2

Thanks to the authors for their rebuttal to my initial review. I am satisfied by the authors comments, explanations and alterations to the manuscript. I am also convinced that the stretchable fibres represent a significant contribution to the field. Therefore, my recommendation is to publish.

Answer. We appreciate the reviewer's careful evaluation of our manuscript.

Reviewer #3

The author has systematically replied to all the messages from the reviewers, and this version can be accepted for publication in Nature Communications.

Answer. We are grateful to the reviewer for their thorough review of our paper.